# Oleate Promotes Triple-Negative Breast Cancer Cell Migration by Enhancing Filopodia Formation through a PLD/Cdc42-Dependent Pathway

**DOI:** 10.3390/ijms25073956

**Published:** 2024-04-02

**Authors:** Zhiqiang Guo, Karl-Frédérik Bergeron, Catherine Mounier

**Affiliations:** Biological Sciences Department, Université du Québec à Montréal (UQAM), Montréal, QC H2X 1Y4, Canada

**Keywords:** oleic acid, triple-negative breast cancer, cell migration, filopodia, Cdc42, Arp2/3 complex, phospholipase D

## Abstract

Breast cancer, particularly triple-negative breast cancer (TNBC), poses a global health challenge. Emerging evidence has established a positive association between elevated levels of stearoyl-CoA desaturase 1 (SCD1) and its product oleate (OA) with cancer development and metastasis. SCD1/OA leads to alterations in migration speed, direction, and cell morphology in TNBC cells, yet the underlying molecular mechanisms remain elusive. To address this gap, we aim to investigate the impact of OA on remodeling the actin structure in TNBC cell lines, and the underlying signaling. Using TNBC cell lines and bioinformatics tools, we show that OA stimulation induces rapid cell membrane ruffling and enhances filopodia formation. OA treatment triggers the subcellular translocation of Arp2/3 complex and Cdc42. Inhibiting Cdc42, not the Arp2/3 complex, effectively abolishes OA-induced filopodia formation and cell migration. Additionally, our findings suggest that phospholipase D is involved in Cdc42-dependent filopodia formation and cell migration. Lastly, the elevated expression of Cdc42 in breast tumor tissues is associated with a lower survival rate in TNBC patients. Our study outlines a new signaling pathway in the OA-induced migration of TNBC cells, via the promotion of Cdc42-dependent filopodia formation, providing a novel insight for therapeutic strategies in TNBC treatment.

## 1. Introduction

Breast cancer is one of the most commonly diagnosed cancers in the world, presenting a significant global health challenge [1]. By immunohistochemistry for the presence of the estrogen receptor (ER), progesterone receptor (PR), and human epidermal growth factor receptor 2 (HER2/neu), breast cancers are commonly classified into different molecular subtypes [2]. Triple-negative breast cancer (TNBC) is defined by the lack of expression of all three receptors [3]. Although TNBC only accounts for 15–20% of breast cancers, this subtype is highly aggressive and prone to metastasis. It has the worst clinical outcomes with greater recurrence and lower overall survival rates. And there are no targeted therapies available yet [4,5]. According to the American Cancer Society, the overall 5-year relative survival rate for American patients with TNBC is 77%, compared with 90% for non-TNBC breast cancer.

Oleic acid (OA) constitutes the most prevalent monounsaturated fatty acid (MUFA) in the human diet, comprising approximately 20% of all dietary fat sources. It is also the principal component of olive oil, accounting for nearly 80% of total oil content [6,7]. The consumption of olive oil is a defining feature of the Mediterranean diet, renowned for its health-promoting attributes and traditionally associated with protective effects against cardiovascular diseases, diabetes, obesity, and cancer [7,8,9,10,11,12]. MUFA, particularly OA, has been the subject of extensive research exploring its potential impact on cancer, including breast cancer [13]. However, the outcomes of these investigations remain inconclusive, yielding both pro-cancer and protective effects [13].

In cancer progression, abnormal cell migration is a pivotal and classical aspect of tumor metastasis, encompassing a multifaceted sequence of events, including tumor cell migration, invasion, intravasation, survival in the circulatory system, extravasation, and regrowth in a new environment [14,15,16]. Unlike normal cell migration, cancer cells can spread and move through various alternative mechanisms, such as amoeboid cell migration, mesenchymal cell migration, and collective cell migration [17]. While migration patterns may differ among tumor microenvironment contexts, the consensus is that the regulation of actin dynamics associated with membrane protrusion serves as a fundamental and shared driver of cell migration [17]. Consequently, the spatial control of the actin cytoskeleton stands as a critical factor in governing cell migration [18,19,20]. Cell membrane ruffling is the formation of actin-rich membrane structures, such as lamellipodia, filopodia, and membrane ruffles, and plays a key role in cell motility [21]. Lamellipodia, large fan-like structures at the leading edge, are the most iconic form of cell protrusion. In motile cells, they adhere weakly to the substratum [22]. They are formed by Arp2/3 complex-dependent actin filaments, which are not only a hallmark of the leading edge, but also the driving force in single migrating cells [23,24,25,26,27]. The Arp2/3 complex, comprising seven subunits and including two actin-related proteins (Arp2 and Arp3), plays a key role in producing branched networks of actin filaments [28]. Ruffles have a similar morphology to lamellipodia, but are non-adherent and often protruding dorsally [29]. Filopodia are finger-like actin-rich membrane protrusions that extend out from the cell edge, mediated by actin-bundling proteins, such as formins, and regulated by various small GTPases of the Rho family, such as Cdc42. They are thought to be explorative, sensing the local environment and controlling directionality, but also maintaining persistence by promoting cell–matrix adhesion at the leading edge [30,31,32,33,34,35,36,37,38]. Our current understanding of the molecular mechanisms of cell migration has been largely influenced by studies in non-cancerous contexts, such as embryonic development, immune response, and tissue repair [39]. However, cancer cells exhibit distinct metabolic reprogramming leading to changes in cell migration and invasion compared to normal cells [40].

Several studies have demonstrated that OA promotes breast cancer cell migration and invasion via GPR40/120, EGFR, and PI3K/Akt-dependent pathways [41]. OA also influences cell adhesion mechanisms, including integrin signaling and focal adhesion kinase activity [42,43,44,45]. In addition, it has been shown that OA is involved in extracellular matrix remodeling by regulating paxillin, matrix metalloproteinases (MMPs), and fibronectin activity [41,44,46,47]. Our previous study revealed initial insights into the link between OA and TNBC cell migration through a PLD-mTOR pathway [48]. However, the precise molecular mechanism involved remains elusive. This prompted us to investigate the impact of OA on remodeling the actin structure in TNBC cell lines, and the underlying signaling pathways. Our research highlighted the pivotal role of Cdc42-dependent filopodia formation in promoting TNBC cell migration, shedding light on a novel avenue for developing strategies for the treatment of TNBC.

## 2. Results

### 2.1. OA-Induced Cell Membrane Ruffling in TNBC Cells

Firstly, we investigated the impact of OA treatment on TNBC cell morphology. OA treatment induced rapid cell membrane ruffling in both MDA-MB-231 and MDA-MB-468 cells (Figure 1A,B). Ruffling area and intensity were both increased, with peak responses at 10 min following the onset of OA treatment (Figure 1C). Moreover, OA-treated TNBC cells displayed prominent filopodia or filopodia-like protrusions (zoomed-in panels in Figure 1A, and Appendix A). A quantitative assessment confirmed the substantial formation of filopodia, with increased length and density, particularly at the 10 min time point (Figure 1D). These findings underscore the dynamic and time-dependent alterations in cell morphology elicited by OA treatment in TNBC cells.

### 2.2. OA-Induced Translocation of Cdc42 and the Arp2/3 Complex in TNBC Cells

Considering the actin-rich cell protrusion changes induced by OA, we delved deeper into the mechanisms underlying these alterations, particularly focusing on the roles of Cdc42 and the Arp2/3 complex. These two molecular players are pivotal in orchestrating cytoskeletal dynamics, including the formation of filopodia and lamellipodia, respectively [23,49]. In both MDA-MB-231 and MDA-MB-468 cells, the localization of Cdc42 was primarily observed in the cytoplasmic and nuclear regions in the absence of OA treatment. Following OA exposure, a distinctive perinuclear distribution pattern of Cdc42 was observed (Figure 2A,B). The quantitative analysis unveiled an increased ratio of cytoplasmic to nuclear fluorescence intensity (Figure 2C), suggesting a dynamic translocation of Cdc42 in response to OA treatment. To locate the Arp2/3 complex, we labeled the TNBC cell with an antibody against its Arp2 subunit. In MDA-MB-231 cells, Arp2 exhibited a nucleus to cytoplasm translocation pattern following OA treatment (Figure 2D). However, this translocation was not as observed in MDA-MB-468 cells. Interestingly, MDA-MB-468 cells rather exhibited a reduction in Arp2 localization to the plasma membrane, with a greater proportion of Arp2 found within the cytoplasm (Figure 2E). Quantification further revealed an increased ratio of cytoplasmic to nuclear fluorescence intensity in MDA-MB-231 cells, but not in MDA-MB-468 cells (Figure 2F). Our findings underscore the involvement of Cdc42 in TNBC cell response to OA treatment, while the involvement of the Arp2/3 complex appears to be contingent on the cell line.

### 2.3. Cdc42 Activity Is Required for OA-Induced Filopodia Formation in TNBC Cells

To further elucidate the involvement of Cdc42 and the Arp2/3 complex in OA-induced cell protrusion formation, we conducted colocalization analyses of the Arp2/3 complex and Cdc42 with cell protrusions in TNBC cell lines. Our results show that both Arp2 and Cdc42 exhibit notable colocalizations with F-actin-rich cell protrusions (Figure 3A). Particularly in MDA-MB-231 cells, Cdc42 exhibited a more pronounced association with filopodia, while Arp2 demonstrated a greater localization in lamellipodia, which is consistent with the established literature [27,38]. Subsequently, we evaluated the impact of disrupting the activity of Cdc42 and the Arp2/3 complex using specific pharmaceutical inhibitors: ML141 and CK666, respectively. MDA-MB-231 and MDA-MB-468 cells were pre-treated with either DMSO (control) or inhibitors for 1 h prior to OA exposure. MTT assays showed no significant cytotoxicity induced by the inhibitors under the conditions used in the experiments (Appendix A). As both Cdc42 and the Arp2/3 complex are critical for the regulation of the cell cytoskeleton, both inhibitors led to a subtle disruption of F-actin structures in the control cells (Figure 3B). In OA-treated cells, ML141 (but not CK666) resulted in a decreased percentage of cells with filopodia (Figure 3B,C). These findings strongly suggest that Cdc42 activity, but not Arp2/3 complex activity, is a requisite factor for the induction of filopodia formation in response to OA treatment.

### 2.4. Cdc42 Activity Is Required for OA-Induced Cell Migration in TNBC Cells

To gain a more comprehensive understanding of the interplay between OA and cell protrusions, and their combined impact on breast cancer cell migration, we conducted wound healing assays in the presence of the two distinct inhibitors for Cdc42 and the Arp2/3 complex (ML141 and CK666). The MTT assay results confirm the absence of significant cell toxicity induced by the inhibitors, as well as the absence of significant proliferation effects induced by OA under the experimental conditions in our wound healing assays (Appendix A). In both MDA-MB-231 and MDA-MB-468 cells, OA treatment promoted wound closure, confirming the pro-migratory effects of OA. Intriguing differences emerged when Cdc42 and Arp2/3 complex inhibitors were introduced. Treatment with ML141 had a significant inhibitory effect on OA-induced wound closure in both TNBC cell lines (Figure 4A,B). However, CK666 demonstrated a variable impact. It effectively inhibited OA-promoted wound closure in MDA-MB-231 cells, but only at a higher concentration (10 μM). In contrast, in MDA-MB-468 cells, CK666 had no effect (Figure 4C,D). Collectively, our results provide compelling evidence that Cdc42 activity is an essential requirement for OA-induced cell migration, while the contribution of Arp2/3 complex activity appears to be context-dependent, contingent on the specific characteristics of the cell line involved.

### 2.5. PLD Is Involved in OA-Induced Filopodia Formation and Cell Migration

Our previous study reported that OA stimulated MDA-MB-231 cell migration in a PLD-dependent pathway, most likely PLD2 [48]. Therefore, we explored the potential involvement of PLD2 in OA-induced filopodia formation and cell migration signaling. Using a highly sensitive and specific phosphatidic acid (PA) sensor known as GFP-PASS, we confirmed that, in both MDA-MB-231 and MDA-MB-468 cells, OA activated PLD activity, as shown by the recruitment of GFP-PASS to the cell membrane (Figure 5A,B) [50]. We then examined the colocalization of Cdc42 and PLD2 in TNBC cells, revealing the colocalization on the cell membrane region of untreated cells. And OA treatment did not influence the expression of Cdc42 and PLD2 (Appendix A), while it prompted the translocation of Cdc42 from the nucleus to the cytoplasm, consequently increasing the degree of colocalization of the two proteins (Figure 5C,D). Much like the effect of OA, treatment with PMA (a strong PLD activator [51,52]) also induced filopodia formation in both MDA-MB-231 and MDA-MB-468 cells (Figure 5E). Our previous research already demonstrated that the inhibition of PLD activity effectively blocked the stimulatory effect of OA on cell migration [48]. Building upon this, we confirmed that the activation of PLD activity by PMA led to a similar stimulating effect on wound healing recovery in TNBC cells (Figure 5F,G). The wound closure-stimulating effect induced by PMA was effectively abrogated by ML141, signifying the indispensable role of Cdc42 activity in PMA-induced cell migration. Taken together, our results suggest that PLD activity is required for OA-induced Cdc42-dependent filopodia formation and cell migration in TNBC cells.

### 2.6. Elevated CDC42 Expression and Its Association with Survival in TNBC Patients

To determine the expression profiles of Cdc42 (gene: CDC42) and Arp2 (gene: ACTR2) across various cancer types, we performed a gene expression analysis utilizing the GEPIA (Gene Expression Profiling Interactive Analysis) platform based on tumor and normal samples from the TCGA and the GTEx databases [53]. Gene expression levels of both CDC42 and ACTR2 were notably elevated in breast cancer tissues compared to their expression in normal tissues, and were ranked among the highest in terms of expression across various cancer types (Appendix A). Further exploration through individual cancer stages revealed that breast cancer patients exhibited heightened expressions of both CDC42 and ACTR2 across all AJCC (American Joint Committee on Cancer) stages, distinguishing them from normal patients (Appendix A). We also investigated the expression of these two genes in 62 breast cancer cell lines, leveraging data from the Human Protein Atlas Database. Interestingly, our analysis revealed that both CDC42 and ACTR2 mRNA expressions were markedly higher in TNBC cell lines in contrast to non-TNBC cell lines (Figure 6A–C).

To assess the potential significance of ACTR2 and CDC42 expressions in breast cancer, we generated Kaplan–Meier survival plots for relapse-free survival (RFS) and distant metastasis-free survival (DMFS) using available gene expression dataset records, spanning a period of up to 180 months. Intriguingly, our analysis unveiled a significant correlation between a high CDC42 expression in tumor tissues and increased mortality rates in TNBC patients compared to all breast cancer patients (Appendix A and Figure 6D). In contrast, the association between high ACTR2 expression and mortality was notably weaker in TNBC patients compared to the broader cohort of breast cancer patients (Appendix A and Figure 6D). The elevated mortality rate observed in the TNBC subset suggests that Cdc42 plays a more crucial role in the development and progression of TNBC compared to non-TNBC types, whereas ACTR2 exhibits the opposite trend. In sum, our bioinformatic analyses bolster our in vitro data supporting a role for Cdc42 in cell migration-related TNBC risk.

## 3. Discussion

Actin cytoskeleton reorganization regulates cell morphological changes, which are intimately linked to cancer cell migration, invasion, and metastasis [54]. Our previous study revealed that OA treatment induced increased cell migration directionality and speed, as well as a more elongated and fibroblast-like shape in MDA-MB-231 cells [48]. Here, we report a novel alteration associated with cell migration induced by OA in TNBC cells. We observed rapid cell membrane ruffling of TNBC cell lines following OA treatment, with an enhanced formation of Cdc42-dependent filopodia (Figure 1 and Figure 3). Aligning with our observations, filopodia-related regulatory mechanisms have been identified in some breast cancer cells. Filopodia and filopodia-like structures are not only prominent features of migrating cancer cells, but also associated with the degree of cancer cell malignancy [55]. For instance, in MCF-7 cells, the oncoprotein HBXIP was found to enhance cell migration by increasing filopodia formation via MEKK2/ERK1/2/Capn4 signaling [56]. In MDA-MB-231 cells, filopodia formation and cell migration were regulated by L-type calcium channels [57] and Cdc42 [35,58]. Furthermore, similar Cdc42-dependent filopodia formation and cell migration observations were made in other cancer types, including colorectal [59], ovarian [60,61], pancreatic [62], and lung cancers [63]. These findings align with our observations of TNBC cell lines, highlighting the crucial roles of filopodia formation, especially through Cdc42 activity, in the context of cancer development and invasion.

Small GTPase Cdc42 is a member of the Rho family and a master regulator of the actin cytoskeleton, controlling cell motility and cell cycle progression [49]. Our results show that ML141, a highly specific Cdc42 inhibitor, efficiently abrogates OA-induced filopodia formation and cell migration in both MDA-MB-231 and MDA-MB-468 cells (Figure 3 and Figure 5). In addition, OA treatment did not change the expression of Cdc42 (Appendix A), while it induced a nucleus to cytoplasm translocation of Cdc42 (Figure 2). This change in spatial distribution could facilitate its functional switch from cell cycle regulation in the nucleus to cytoskeleton regulation in the cytoplasmic region. Furthermore, as a small GTPase, Cdc42 is activated through the exchange of GDP for GTP. This reaction is mediated by guanine nucleotide exchange factors (GEFs), which catalyze the release of GDP and loading of GTP [64]. Most Rho-GEFs localize either in the cytoplasm or in the plasma membrane (PM), and only a few of them are detected in the nucleus [65]. Therefore, the cytoplasmic distribution of Cdc42 (induced by OA treatment) increases its likeliness of being activated by GEFs. Although it is still unclear how OA activates Cdc42, it is well-recognized that Cdc42 can be activated by a number of cell surface receptors, such as G protein coupled receptors (GPCRs), receptor tyrosine kinases (RTKs), and integrin receptors, which converge on Cdc42 by activating specific GEFs [64]. OA is known to be involved in activating GPR40/120 [13,41,66], insulin receptor (a member of RTK family) [67], and integrin receptor signaling [13,42,43], which, in turn, can potentially activate Cdc42 via downstream GEFs. Moreover, lipid modifications play an important role in the regulation of Cdc42 activity [68,69]. The C-terminal region of Cdc42 contains a CAAX box that is a site for post-translational lipid modification, which regulates its localization and activity [68]. As a possible lipid moiety, OA might also directly influence the localization and activation of Cdc42 by changing its lipidation state.

Using bioinformatics tools, we found elevated expressions of CDC42 and ACTR2 in breast cancer. In TNBC, however, the high expression of CDC42 in the primary tumor was clearly correlated with cancer-related death. Interestingly, this association was even higher in TNBC patients (Figure 6). Although relatively few Cdc42 oncogenic mutations have been reported in cancer [49,70], the overexpression of Cdc42 is observed in several types of cancers, such as breast [32,71,72,73], colorectal [74], esophageal [75], gastric [74], lung [76,77,78], melanoma [79], ovarian [80,81], pancreatic [82], and testicular cancers [83]. Some studies also found the overexpression of Cdc42 to be positively correlated with a poor prognosis [79,81,82,83]. This further supports Cdc42 as a potential therapeutic target for cancer treatment, especially in TNBC, which does not respond to most therapies.

The elevated expression and activity of PLD, especially PLD2, have been detected in various human cancer tissues and cells, including breast cancer [84,85,86]. Our previous study also reported that PLD expression was associated with an increased proportion of metastasis-related deaths among TNBC patients. We also showed that PLD activity was involved in OA-induced MDA-MB-231 cell migration and invasion [48]. Therefore, we further investigated if PLD was involved in OA-induced filopodia formation signaling. Here, we confirmed that OA could activate PLD activity in TNBC cell lines (Figure 5A). Although, we could not distinguish which isoform of PLD was activated due to our methodology. Several lines of evidence have suggested that PLD2 is the isoform stimulated by OA. For instance, Kim et al. reported that OA selectively stimulated the enzymatic activity of PLD2, but not of PLD1, in vitro [87]. PLD activity was highly stimulated by OA in Jurkat T cells (only expressing PLD2), but not in HL-60 cells (only expressing PLD1) [88,89]. In RBL-2H3 mast cells, OA stimulated PLD activity only when PLD2, but not PLD1, was overexpressed [90]. These all support the idea that OA can activate PLD2 in TNBC cells. Next, we further explored if the activation of PLD could trigger Cdc42-dependent filopodia formation and cell migration signaling. Our results show that, much like OA, the activation of PLD by PMA induces filopodia formation as well as cell migration in TNBC cells. This effect on migration was dependent upon Cdc42 activity (Figure 5G). In line with a role in cell migration, PLD2 was found to be frequently localized to the leading edge of motile cells in membrane ruffles [91,92]. Previous studies have shown that the elevated expression of PLD2 substantially increases the length of cell protrusions, while a catalytically inactive PLD2 mutant abolishes them [93]. Therefore, we investigated the potential interaction between Cdc42 and PLD2. Our results reveal that Cdc42 and PLD2 colocalize and OA treatment increases their degree of colocalization (Figure 5C,D), suggesting a direct interaction. PLD2 possesses a unique GEF feature. It was reported that PLD2 potently stimulated GDP-GTP exchange on Rac2, a Rho family member involved in filopodia formation [34,94], just like Cdc42. There are two CRIB (Cdc42 and Rac interactive binding) motifs in and around the PH domain of PLD2 [95], raising the possibility that PLD2 can act as a GEF for Cdc42. However, further research is required to clarify the potential connection between PLD2 and Cdc42.

In addition to Cdc42, the Arp2/3 complex has been implicated in the formation of cell protrusions and in the cell migration of motile cells. However, there remains some controversy whether it is indispensable in filopodia formation. In our study, two TNBC cell lines showed different responses to Arp2/3 complex activity. MDA-MB-468 cells showed no response to CK666, a Arp2/3 complex inhibitor, in OA-induced filopodia formation and migration (Figure 3C and Figure 4D, respectively), while in MDA-MB-231 cells, OA induced a nucleus to cytoplasm translocation of the Arp2/3 complex, and a high concentration of CK666 inhibited OA-stimulated wound closure (Figure 2D–F and Figure 4C,D). These results support the involvement of the Arp2/3 complex in OA-induced filopodia formation and cell migration in MDA-MB-231 cells. Although, perhaps surprisingly, CK666 did not decrease the percentage of cells with filopodia (Figure 3C). We assume this difference was caused by different filopodia formation mechanisms. There are two alternative models of filopodia initiation: the convergent elongation model and the tip nucleation model. In the convergent elongation model, filopodia emerge from a lamellipodial actin meshwork that is assembled through Arp2/3 complex-mediated nucleation [96]. In contrast, the tip nucleation model proposes that filopodia are able to self-assemble directly via the action of formins on the plasma membrane [96], without the need for an Arp2/3 complex-dependent lamellipodial core. It is likely that both models of filopodia formation exist in MDA-MB-231 cells, while the self-assembly model appears more dominant in MDA-MB-468 cells. In addition, lamellipodia provide the driving force for forward movement, while filopodia contribute to the sensing of the microenvironment, allowing the cell to navigate in a directed manner [18,23,30,34,38]. The coordination and interplay between filopodia and lamellipodia during cell migration could contribute to a higher migration speed compared to single protrusion-based movements. This might partly account for why MDA-MB-231 cells demonstrated a higher basal migration speed, both with and without OA stimulation, compared with MDA-MB-468 cells (Figure 4).

In summary, our study unveils a novel signaling pathway that orchestrates the OA-induced migration of TNBC cells, highlighting the pivotal role of the PLD/CDC42 axis in facilitating filopodia formation and subsequent cell motility. Our findings shed light on the intricate molecular mechanisms underlying TNBC metastasis, providing valuable insights for the development of innovative therapeutic strategies. Additional studies will be needed to explore the involvement of the Arp2/3 complex in the context of TNBC migration. Elucidating the role of the Arp2/3 complex in filopodia signaling could also provide a more comprehensive understanding of actin dynamics during cell migration and invasion. Additionally, investigating the GEF functions of PLD2 and how they might contribute to the activation of CDC42 has the potential to offer valuable insights into the intricacies of this signaling pathway. These insights will bring us closer to the prospect of developing innovative and precise therapeutic strategies for combatting TNBC metastasis, addressing a critical aspect of cancer progression that has significant implications for patient outcomes and overall survival.

## 4. Materials and Methods

### 4.1. Materials

MDA-MB-231 cells were kindly gifted by Dr. Jean-Jacques Lebrun (McGill University, Montreal, QC, Canada). MDA-MB-468 cells were kindly gifted by Dr. Borhane Annabi (UQAM, Montreal, QC, Canada). PASS biosensor plasmids (GFP-PASS and RFP-PASS), kindly gifted by Dr. David N. Brindley (University of Alberta, Canada), originated from Dr. Guanwei Du (The University of Texas Health Science Center at Houston, TX, USA). Eagle’s minimum essential medium (EMEM, #320-005-CL) and Dulbecco’s modification eagle’s medium (DMEM, #319-005-CL) were purchased from Wisent. PLD2 antibody (7E4D9, #MA5-31854), Cdc42 antibody (#PA1-092), Arp2 antibody (5H2L7, #703394), and Lipofectamine 3000 Transfection Reagent were purchased from Invitrogen Thermo Fisher Scientific. PLD2 antibody (E1Y9G, #13904), HRP-conjugated anti-rabbit IgG (#7074), anti-rabbit IgG (H+L), F(ab’)2 Fragment (Alexa Fluor 488 Conjugate, #4412), and anti-mouse IgG (H+L), F(ab’)2 Fragment (Alexa Fluor 647 Conjugate, #4410) were purchased from Cell Signaling Technology. Bovine serum albumin (BSA)-conjugated oleic acid (OA; #O-3008), ML141 (SML0407), CK666 (SML0006), and Phorbol 12-myristate 13-acetate (PMA, P8139) were purchased from Sigma-Aldrich (MilliporeSigma Canada Ltd., Oakville, ON, Canada).

### 4.2. Cell Culture and Transfection

MDA-MB-231 cells were cultured in EMEM. MDA-MB-468 cells were cultured in DMEM. Both culture media were supplemented with 10% fetal bovine serum (FBS, Gibco heat inactivated, Thermo Fisher Scientific, Waltham, MA, USA), 500 U/mL of penicillin, and 500 μg/mL streptomycin (LT Gibco, #15070063). Cells were maintained in a 5% CO_2_ incubator at 37 °C. The cells used in these experiments were between passages 5 and 25. Adherent cells were detached using 0.25% Trypsin-EDTA (Gibco, 25200056). Cells were transfected with Lipofectamin 3000 following the manufacturer’s instructions.

For treatments of cells, BSA-conjugated OA was used at 50–100 μM with fatty acid free BSA as the control. Cells were also incubated with Cdc42 inhibitor ML141 and Arp2/3 complex inhibitor CK666 at 5–20 μM for 1 h before treatment or overnight with PMA at 10 ng/mL.

### 4.3. Immunofluorescence Staining and F-Actin Staining

MDA-MB-231 and MDA-MB-468 cells were seeded at a density of around 50–70% confluency in 24-well plates on sterilized coverslips. After 24 h of incubation and following treatments, the cells were washed with ice-cold phosphate-buffered saline (PBS) buffer, pH 7.4, fixed with 4% paraformaldehyde in PBS for 20 min and washed again. The fixed cells were permeabilized for 10 min with 0.1% Triton X-100 and blocked with 1% BSA for 1 h at room temperature. The cells were then incubated for 20 min with 50 μg/mL of Phalloidin-TRITC Reagent (P1951, Sigma-Aldrich) and 1 μg/mL of nuclear counterstain 4′,6-diamidino-2-phenylindole (DAPI) in PBS, according to the manufacturer’s instructions. For immunofluorescence staining, we followed the Immunofluorescence Protocols Guidelines from Thermo Fisher Scientific. Briefly speaking, the cells were incubated overnight at 4 °C with primary antibody. After washing with PBS, the cells were then incubated in properly diluted secondary Alexa Fluor antibodies in the dark for 1 h at room temperature. Lastly, the coverslips were placed cell-side down onto a drop of mounting medium (90% glycerol in PBS) and sealed with the microscope slide using clear nailpolish. The samples were stored at 4 °C before imaging.

### 4.4. Confocal Microscopy and Image Analysis

Fluorescent images were obtained with a Nikon A1 plus inverted confocal microscope (63 × NA oil objective). Image processing, including Z-stacking, was performed using ImageJ Fiji software (OpenJDK v13.0.6).

Cell membrane ruffling assay was performed by Ruffle Analysis Macro in ImageJ Fiji, following instructions [97]. Cell numbers and nuclei mid-point offsets were manually corrected.

For the filopodia quantification analysis, at least 60 random cells of each condition from 3 independent experiments were analyzed. The FiloQuant plugin v1.1 [55] for ImageJ Fiji Macro was utilized to analyze filopodia number, density, and length. For MDA-MB-231 cells, FiloQuant single-image analysis was used to detect and measure the length and the number of filopodia. For MDA-MB-468 cells, only cell edge regions without cell–cell contact were randomly chosen for this analysis. The cell edge length was manually corrected using the Free Hand Line tool and Multiple Point tool. Filopodia density was defined as a ratio of the number of detected filopodia to cell edge length.

Fluorescence intensity analyses were performed using a line-intensity histogram from a selected line spanning the cell using ImageJ Fiji, as modified from Lu et al. [50].

Colocalization analyses were performed by Coloc2 (v3.0.6) plugin in ImageJ Fiji. Pearson’s coefficient value was used as a colocalization index.

### 4.5. Wound Healing Assay

MDA-MB-231 and MDA-MB-468 cells were seeded in 24-well plates at a density of 1 × 10^5^ cells per well. Having reached around 80% confluency, the cells were treated for 24 h with OA or PMA. For inhibitors analysis, the cells were pretreated with inhibitors 4 h before OA or PMA treatments. The confluent cell monolayer was scratched with P-200 tips. The wound recovery was monitored under 5% CO_2_ and at 37 °C. Images were acquired by a Nikon Eclipse Ti inverted microscope. The wound area at each acquisition was measured by the Wound Healing-Size Tool Macro plugin in Image J Fiji [98]. Wound closure corresponds to the shrinking wound surface area relative to the initial area.

### 4.6. Cell Viability Assay

MDA-MB-231 and MDA-MB-468 cell viabilities were evaluated with an MTT assay modified from the MTT Assay Protocol from Millipore Sigma. Cells were seeded at a density of about 80% confluency in 96-well plate. After the incubation with treatments, 10 μL of the MTT labeling reagent (final concentration of 0.5 mg/mL, 475989, Sigma-Aldrich) was added to each well and incubated for 4 h in a cell incubator. The formazan crystals were dissolved in lysis solution (10% NP-40, 10 mM HCl) overnight at 37 °C, and absorbance was measured at 570 nm with correction at 690 nm with a BioTek Eon Microplate Spectrophotometer.

### 4.7. Bioinformatic Analyses

Gene expression analyses of human CDC42 and ACTR2 across various human cancer types were performed using GEPIA (Gene Expression Profiling Interactive Analysis, http://gepia.cancer-pku.cn (accessed on 10 September 2023) [53]. CDC42 and ACTR2 gene expression analyses in human breast cancer stages were performed using UALCAN (University of ALabama at Birmingham CANcer data analysis Portal, https://ualcan.path.uab.edu (accessed on 9 September 2023 [99]). Gene expressions in breast cancer cell lines were based on the Human Protein Atlas database (https://www.proteinatlas.org (accessed on 9 September 2023). Kaplan–Meier survival plots were generated using the Kaplan–Meier Plotter (https://kmplot.com/analysis/ (accessed on 9 September 2023). Breast cancer patients were separated into 2 groups based on expressions of ACTR2 (probe 200729_s_at) and CDC42 (probe 210232_at) in primary tumors over a period up of to 180 months using the “best cutoff” option.

### 4.8. Western Blotting

MDA-MB-231/-468 cells were lysed in RIPA buffer containing protease and phosphatase inhibitors (Sigma, #P8340; #P0044). After centrifugation, proteins were recovered in the supernatant. Protein concentration was determined by the Bradford protein assay (Bio-Rad, #5000006). Proteins were separated on SDS-PAGE transferred to PVDF membranes. Primary antibodies PLD2 (1:1000; Cell Signaling, #13904) and Cdc42 (1:500; Invitrogen, #PA1-092) were used. HRP-conjugated anti-rabbit IgG (1:2000: Cell Signaling, #7074) was used as a secondary antibody. Signals were revealed using the ECL substrate (Millipore, #WBKLS0100). To normalize and verify protein amounts equally, membranes were finally stained with amido black solution (0.25% amido black, 45% MeOH, 45% ddH2O, 10% glacial HOAc) and de-stained with the same solution without dye.

### 4.9. Statistical Analyses

Statistical analyses were carried out using GraphPad Prism version 8.0. The significance of differences between groups was tested using Student’s *t*-test. Differences were considered significant when *p*-values were <0.05.

## Figures and Tables

**Figure 1 ijms-25-03956-f001:**
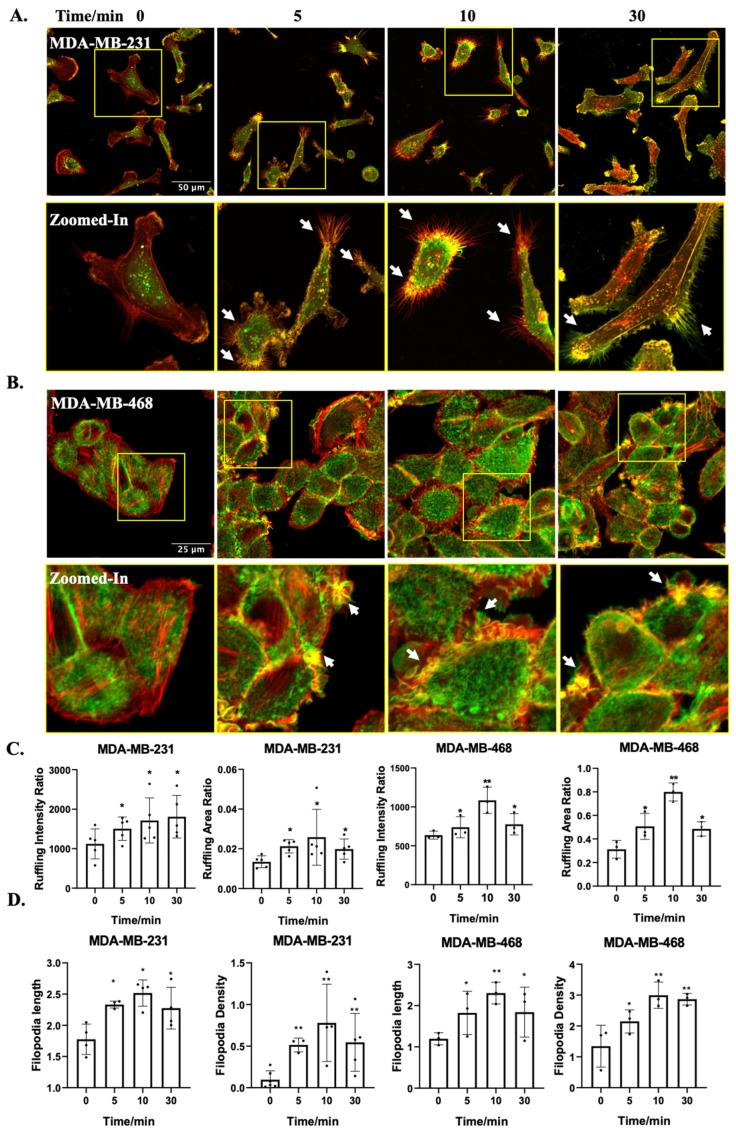
OA-induced morphological changes in TNBC cell lines. (**A**,**B**) Representative fluorescence microscopy images of MDA-MB-231 and MDA-MB-468 cells. Cells were treated with 100 μM OA for the indicated times (5, 10, and 30 min), followed by Phalloidin-TRITC staining. Z-stack projection is pseudo-colored: bottom-cell F-actin in red, and ruffles (upper-cell F-actin) in green. The zoomed in images show the yellow squares in the images above. The scale bar depicts in the leftmost image applies uniformly to all images within the same set. The white arrows point to representative filopodia structures. (**C**) Quantitative assessment of the extent of dorsal ruffling induced by OA treatment. (**D**) Quantification of OA-induced filopodia formation from Appendix A. Data were acquired from at least 60 cells in 3 independent experiments. Statistical significance was evaluated via the Student’s *t*-test (* *p* < 0.05, ** *p* < 0.01).

**Figure 2 ijms-25-03956-f002:**
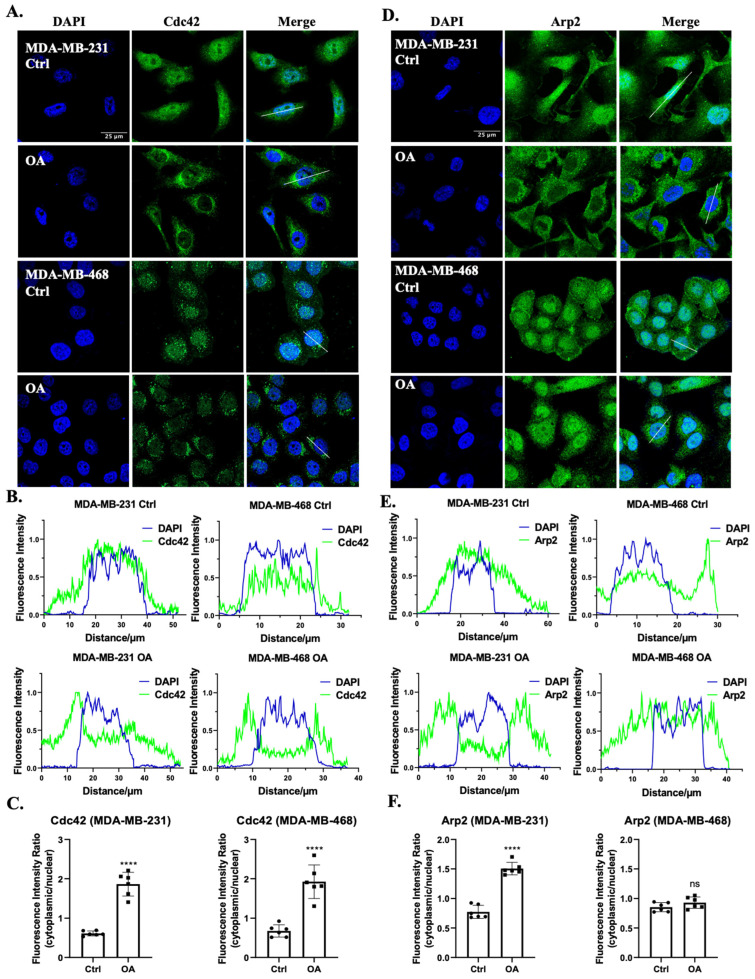
Subcellular localization of Cdc42 and the Arp2/3 complex in response to OA treatment in TNBC cells. (**A**,**D**) Representative immunofluorescence confocal microscopy images of Cdc42 (**A**) and Arp2 (**D**) in MDA-MB-231 and MDA-MB-468 cells. Cells were treated with BSA (Ctrl) or 100 μM OA for 10 min, then stained with specific human Cdc42/Arp2 antibodies (green) and counterstained with DAPI (blue). The scale bar shown in the first image is applicable to all images within the same panel. (**B**,**E**) Fluorescence intensity histograms of DAPI (blue) and Cdc42 (**B**)/Arp2 (**E**) (green) from the lines shown in (**A**,**D**) (see merge columns). (**C**,**F**) Quantitative assessment of Cdc42 and Arp2 fluorescence intensities within the nucleus relative to the cytoplasmic region. Statistical significance was determined using the Student’s *t*-test (ns, not significant; **** *p* < 0.0001).

**Figure 3 ijms-25-03956-f003:**
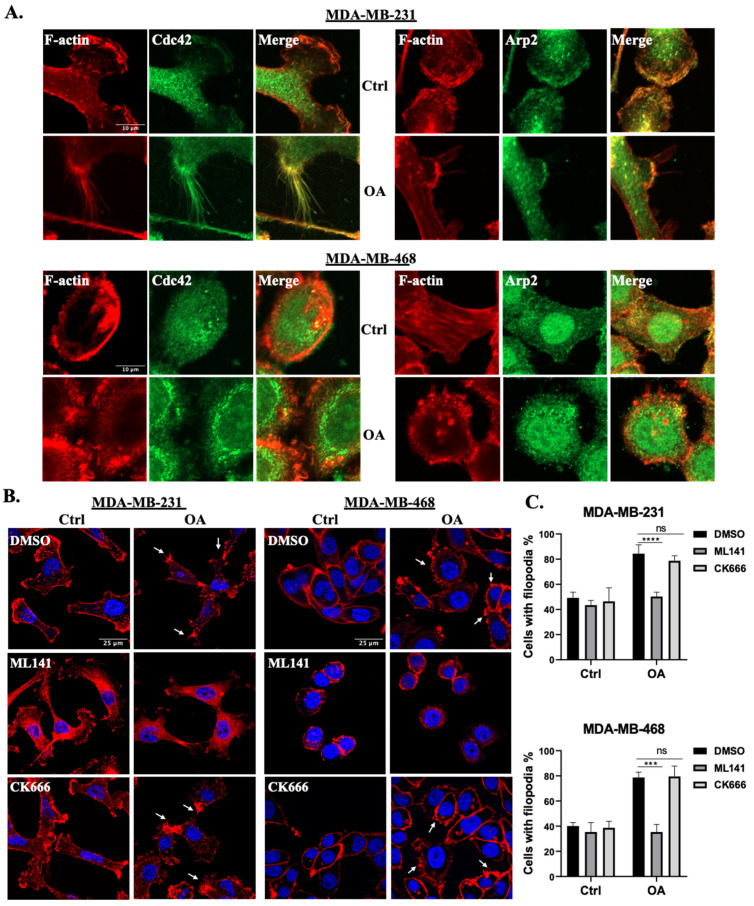
OA promotes filopodia formation in TNBC cells via Cdc42 activation. (**A**) Representative immunofluorescence confocal microscopy images of the localization of Cdc42 and the Arp2/3 complex in MDA-MB-231 and MDA-MB-468 cells. Cells were treated with BSA (Ctrl) or 100 μM OA for 10 min, and subsequently stained with specific Cdc42/Arp2 antibodies (green) and counterstained with Phalloidin-TRITC (red). (**B**) Effects of Cdc42 inhibition (ML141) and Arp2/3 complex inhibition (CK666) on OA-induced cell protrusions. MDA-MB-231 and MDA-MB-468 cells were pretreated with either DMSO (Ctrl), 20 μM ML141 or CK666 for 1 h, and subsequently exposed to BSA (Ctrl) or 100 μM OA for 10 min. The cells were then stained with DAPI (blue) and Phalloidin-TRITC (red). White arrows indicate representative filopodia structures. The scale bars featured in the first image apply uniformly to all images within the same set. (**C**) Images in (**B**) were quantified by counting the percentage of cells presenting filopodia. Data are aggregates of three experiments with at least 50 cells per experiment. Statistical significance was determined using the Student’s *t*-test (ns, not significant; *** *p* < 0.001, **** *p* < 0.0001).

**Figure 4 ijms-25-03956-f004:**
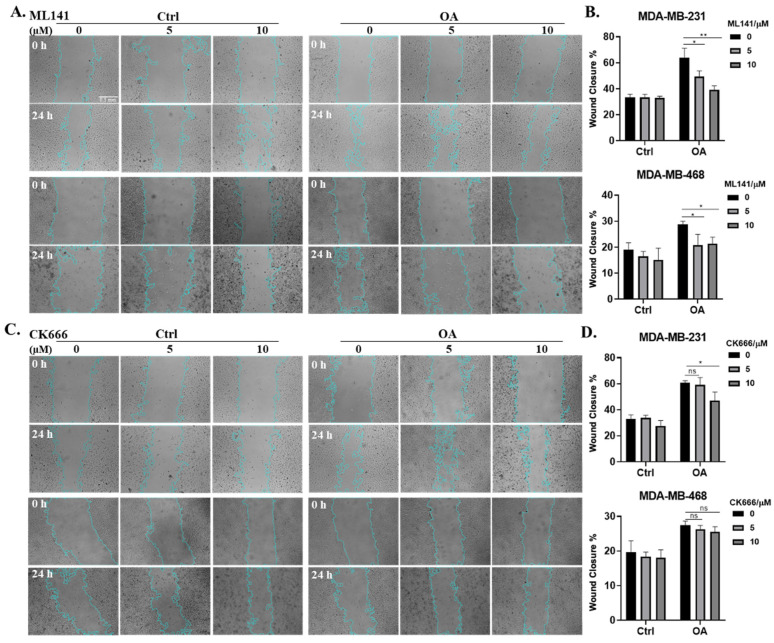
Effects of Cdc42 and Arp2/3 complex inhibitions on OA-induced TNBC cell migration. (**A**,**C**) Representative light-microscopy images of wound healing assays for MDA-MB-231 and MDA-MB-468 cells. Wound healing was evaluated over a 24 h period following BSA (Ctrl) or 50 μM OA treatment in the presence of indicated concentrations of inhibitors (0 μM: DMSO; 5/10 μM: ML141 (**A**)/CK666 (**C**)). The scale bar featured in the first image applies uniformly to all images in the figure. (**B**,**D**) Quantification of wound closure in (**A**,**C**). The average (±standard deviation) percentage of wound closure was calculated from 3 independent experiments. Significance was determined by comparing the inhibitor subgroups (5/10 μM) with the control subgroups (0 μM) within the OA treatment groups using the Student’s *t*-test (ns, not significant, * *p* < 0.05, ** *p* < 0.01).

**Figure 5 ijms-25-03956-f005:**
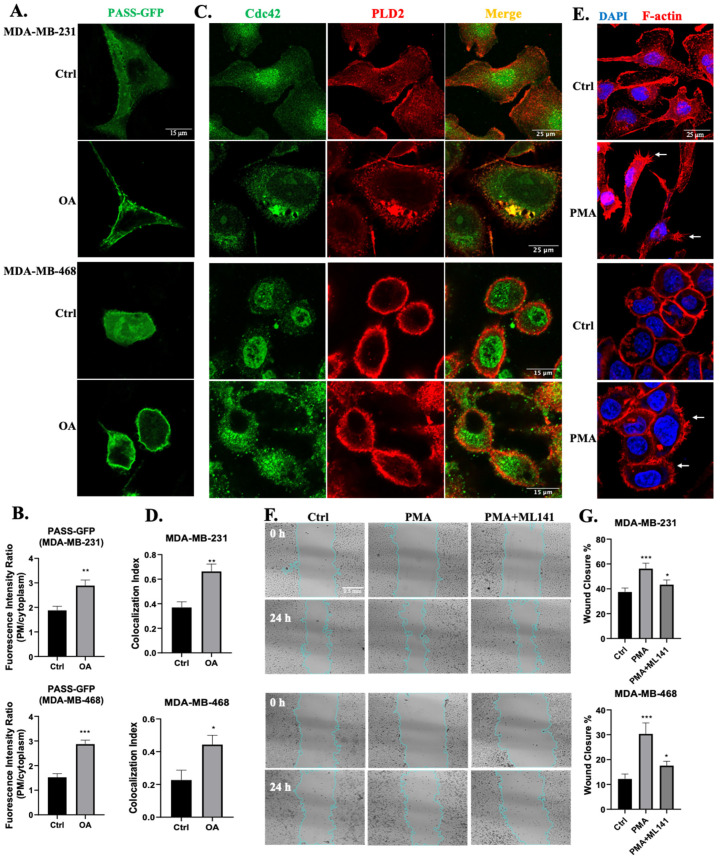
Involvement of PLD in OA-induced filopodia formation and TNBC cell migration. (**A**) Representative confocal microscopy images of PASS-GFP in response to BSA (Ctrl) or 100 μM OA treatment for 10 min in MDA-MB-231 and MDA-MB-468 cells. (**B**) Quantification of the fluorescence intensity of PASS-GFP on the plasma membrane (PM) relative to the cytoplasm. (**C**) Colocalization of Cdc42 and PLD2 in TNBC cell lines. Cells were treated with BSA (Ctrl) or 100 μM OA for 10 min, and subsequently stained with PLD2 antibody (red) and Cdc42 antibody (green). (**D**) Quantification of Cdc42 and PLD2 colocalization using Pearson’s coefficient. (**E**) Induction of filopodia formation in TNBC cells by PMA. Cells were treated with BSA (Ctrl) or 10 ng/mL PMA for 10 min, and subsequently stained with DAPI (blue) and Phalloidin-TRITC (red). (**F**) Representative light-microscopy images of wound healing assays for MDA-MB-231 and MDA-MB-468 cells. Wound healing was evaluated over a 24 h period following treatment with BSA (Ctrl) or in the presence of PMA (10 ng/mL) and Cdc42 inhibitor ML141 (5 μM). (**G**) Quantification of wound closure in (**F**). The average (±standard deviation) percentage of wound closure was calculated from 3 independent experiments. Statistical significance was determined via the Student’s *t*-test (* *p* < 0.05, ** *p* < 0.01, *** *p* < 0.001). The scale bars featured apply uniformly to all images within their sets.

**Figure 6 ijms-25-03956-f006:**
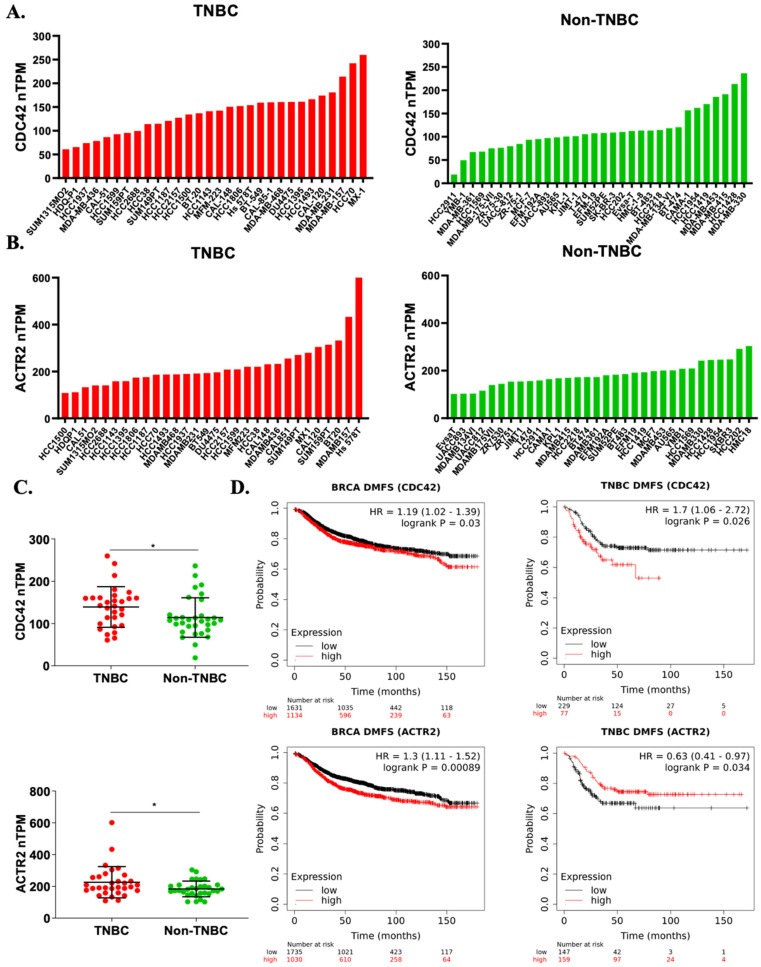
High CDC42 expression correlates with TNBC risks. (**A**–**C**) Gene expression profiles of human CDC42 and ACTR2 in a panel of breast cancer cell lines (from the Human Protein Atlas). Statistical significance was assessed by the Student’s *t*-test (* *p* < 0.05). (**D**) Kaplan–Meier survival plots of distant metastasis-free survival (DMFS) in breast cancer patients (all patients and TNBC subset) over a span of up to 180 months, based on CDC42 and ACTR2 expressions. “HR” stands for “hazard ratio”, and “log-rank P” refers to the *p*-value obtained from a log-rank test.

## Data Availability

All data sources referenced in the manuscript have been appropriately cited. Additional original data can be made available upon request by contacting us directly.

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
