# Peer review of "Oleate Promotes Triple-Negative Breast Cancer Cell Migration by Enhancing Filopodia Formation through a PLD/Cdc42-Dependent Pathway"

_ijms, 2024, doi:10.3390/ijms25073956_

Round 1

Reviewer 1 Report

Comments and Suggestions for Authors

Oleate Promotes Triple Negative Breast Cancer Cell Migration 2 by Enhancing Filopodia Formation through a PLD/Cdc42‐de‐3 pendent Pathway.

‐ The used OA, oleic acid‐conjugated with BSA does not represent the olive oil oleic acid since the total acids in EVOO reasonable quality is less than 0.8%. More than 80% of EVOO fats are oleic acid triglycerides, an acylated oleic acid with glycerol. Therefore, the manuscript in vitro data is not expressing the monounsaturated fatty acid oleic acid in EVOO. To correct this, authors should test the oleic acid triglycerides from EVOO and see if they will see the same effects otherwise the study is not really related to olive oil.

‐ In lines 64‐69, the authors stated “Filopodia are finger‐64 like actin‐rich membrane protrusions that extend out from the cell edge, mediated by ac‐65 tin‐bundling proteins such as formins, and regulated by various small GTPases
of the Rho 66 family such as Cdc42. They are thought to be explorative, sensing the local environment, 67 and controlling directionality but also maintaining persistence by promoting cell‐matrix 68 adhesiveness at the leading
edge [30‐38]” How come inducing filopodia will promote migration although the original function is to promote cell‐matrix adhesiveness?

‐ Authors relied entirely on the use of wound‐healing/scratch assay. This model is actually 50% migration and 50% proliferation because cells proliferation will also contribute to wound closure and authors relied on the wound‐healing distance rather than counting the migrated cells within the wound. Authors
may need to find a second validating model for migration, like spheroids and/or others to confirm effects on migration.

‐ Have the authors observed the filopodia‐promoting effect on a non‐tumorigenic breast cells like MCF‐10 or MCF‐12?

‐ In Figure 4, the legend is messed. “Would” should be “Wound”, “5 and 10” should be “Control and 50 μM OA”.

‐ Testing OA at concentrations of 50 and 100 μM is not really physiologically relevant nor practical. Have the investigators tested this effect at lower concentrations, more physiologically realistic?

‐ I recommend the authors look at some motility markers like FAK or others.
‐ I do not see the part concerning PMA relevant to the current study since this was published earlier by the same group and therefore Figures 5E, F, and G are not really relevant and not necessary.

‐ I do not see the link nor any current study data showing the role of PLD2.
‐ Authors should show the molecular expression levels of PLD2, CDC42 in both used TNBC cell lines and the effect of OA treatment on their expressions.

‐Have the authors assessed the OA treatment effects on the used TNBC cells proliferation?

‐Authors should do a better job in linking ACTR2 with the current study. It came suddenly in line 227 without any introduction, certainly it is associated with motility but this should be detailed.

‐ The bioinformatics part is very diffused and unfocused. It should focus on TNBC rather than the current wide approach.

Reviewer 2 Report

Comments and Suggestions for Authors

I reviewed the manuscript titled “Oleate Promotes Triple Negative Breast Cancer Cell Migration by Enhancing Filopodia Formation through a PLD/Cdc42-dependent Pathway”

The manuscript is well written and meet the standards of IJMS.

Abstract

Authors must revise the abstract based on background of the study. For example, Our recent study revealed ……. Should not be in abstract section. Instead, authors should provide it as background.

There are  no clear objectives. Results should be explored clearly. Keywords should be revised.

The introduction is appropriate.

Methodology

4.3. Immunofluorescence staining and F-actin staining: add citation.

4.6. Cell viability assay: add citation.

Results : this section is appropriate; however:

Lines  83-85: not required in results section

Figure 1C and D: improve the quality of the figure

Figure 4 b and D: improve the quality of the figure

Figure 6: improve the quality of the figure

Discussion should be improved with supported literature. I suggest adding more recent literature.

References should be cross-checked for correctness.

Authors could provide a graphical abstract for better understanding by the readers.

After careful review, I am pleased to recommend the manuscript for minor revisions as it exhibits no major scientific flaws. The quality of the manuscript is commendable, with no significant issues noted.

Comments on the Quality of English Language

I reviewed the manuscript titled “Oleate Promotes Triple Negative Breast Cancer Cell Migration by Enhancing Filopodia Formation through a PLD/Cdc42-dependent Pathway”

The manuscript is well written and meet the standards of IJMS.

Abstract

Authors must revise the abstract based on background of the study. For example, Our recent study revealed ……. Should not be in abstract section. Instead, authors should provide it as background.

There are  no clear objectives. Results should be explored clearly. Keywords should be revised.

The introduction is appropriate.

Methodology

4.3. Immunofluorescence staining and F-actin staining: add citation.

4.6. Cell viability assay: add citation.

Results : this section is appropriate; however:

Lines  83-85: not required in results section

Figure 1C and D: improve the quality of the figure

Figure 4 b and D: improve the quality of the figure

Figure 6: improve the quality of the figure

Discussion should be improved with supported literature. I suggest adding more recent literature.

References should be cross-checked for correctness.

Authors could provide a graphical abstract for better understanding by the readers.

After careful review, I am pleased to recommend the manuscript for minor revisions as it exhibits no major scientific flaws. The quality of the manuscript is commendable, with no significant issues noted.
